# Anserine, a Histidine-Containing Dipeptide, Suppresses Pressure Overload-Induced Systolic Dysfunction by Inhibiting Histone Acetyltransferase Activity of p300 in Mice

**DOI:** 10.3390/ijms25042344

**Published:** 2024-02-16

**Authors:** Yoichi Sunagawa, Ryosuke Tsukabe, Yudai Irokawa, Masafumi Funamoto, Yuto Suzuki, Miho Yamada, Satoshi Shimizu, Yasufumi Katanasaka, Toshihide Hamabe-Horiike, Yuto Kawase, Ryuya Naruta, Kana Shimizu, Kiyoshi Mori, Ryota Hosomi, Maki Komiyama, Koji Hasegawa, Tatsuya Morimoto

**Affiliations:** 1Division of Molecular Medicine, School of Pharmaceutical Sciences, University of Shizuoka, Shizuoka 422-8526, Japan; y.sunagawa@u-shizuoka-ken.ac.jp (Y.S.); m18156@u-shizuoka-ken.ac.jp (R.T.); funamoto@u-shizuoka-ken.ac.jp (M.F.); s.shimizu@u-shizuoka-ken.ac.jp (S.S.); katana@u-shizuoka-ken.ac.jp (Y.K.); t.hamabe@u-shizuoka-ken.ac.jp (T.H.-H.); koj@kuhp.kyoto-u.ac.jp (K.H.); 2Division of Translational Research, National Hospital Organization Kyoto Medical Center, Kyoto 612-8555, Japan; 3Shizuoka General Hospital, Shizuoka 420-8527, Japan; mori@u-shizuoka-ken.ac.jp; 4Department of Pharmacology, Institute of Biomedical Sciences, Tokushima University Graduate School, Tokushima 770-8503, Japan; 5Graduate School of Public Health, Shizuoka Graduate University of Public Health, Shizuoka 420-0881, Japan; 6Department of Molecular and Clinical Pharmacology, School of Pharmaceutical Sciences, University of Shizuoka, Shizuoka 422-8526, Japan; 7Laboratory of Food and Nutritional Sciences, Faculty of Chemistry, Materials and Bioengineering, Kansai University, Osaka 564-8680, Japan; hryotan@kansai-u.ac.jp

**Keywords:** anserine, p300, histone acetyltransferase activity, heart failure, pressure overload, cardiomyocyte hypertrophy

## Abstract

Anserine, an imidazole dipeptide, is present in the muscles of birds and fish and has various bioactivities, such as anti-inflammatory and anti-fatigue effects. However, the effect of anserine on the development of heart failure remains unknown. We cultured primary cardiomyocytes with 0.03 mM to 10 mM anserine and stimulated them with phenylephrine for 48 h. Anserine significantly suppressed the phenylephrine-induced increases in cardiomyocyte hypertrophy, ANF and BNP mRNA levels, and histone H3K9 acetylation. An in vitro histone acetyltransferase (HAT) assay showed that anserine directly suppressed p300-HAT activity with an IC_50_ of 1.87 mM. Subsequently, 8-week-old male C57BL/6J mice were subjected to transverse aortic constriction (TAC) and were randomly assigned to receive daily oral treatment with anserine-containing material, Marine Active^®^ (60 or 200 mg/kg anserine) or vehicle for 8 weeks. Echocardiography revealed that anserine 200 mg/kg significantly prevented the TAC-induced increase in left ventricular posterior wall thickness and the decrease in left ventricular fractional shortening. Moreover, anserine significantly suppressed the TAC-induced acetylation of histone H3K9. These results indicate that anserine suppresses TAC-induced systolic dysfunction, at least in part, by inhibiting p300-HAT activity. Anserine may be used as a pharmacological agent for human heart failure therapy.

## 1. Introduction

Heart failure results in lowered cardiac output and various symptoms such as fatigue, decreased exercise tolerance, dyspnea and edema. Persistent stress, such as hypertension, induces compensatory activation of the sympathetic nervous system and the renin–angiotensin–aldosterone system to maintain cardiac output [1]. Prolonged preloading and after-loading of the heart due to the activation of these neuronal-humoral factors leads to cardiac remodeling, including ventricular hypertrophy and enlargement, as well as myocardial fibrosis, which further exacerbates the pathophysiology of chronic heart failure with reduced systolic function [2,3].

Current drug therapies for heart failure are effective; however, radical treatment is required for a complete cure. The identification of more specific therapeutic targets for the disease (e.g., cardiac remodeling, inflammation, fibrosis and diastolic dysfunction) may be more effective than the current approaches [4]. A number of studies have shown that p300 histone acetyltransferase (HAT), a transcriptional coactivator in the nuclei of cardiomyocytes, is implicated in pathological cardiomyocyte hypertrophy and the development of cardiac dysfunction [5,6]. Balasubramanyam et al. reported that curcumin ((1E,6E)-1,7-bis(4-hydroxy-3-methoxyphenyl) hepta-1,6-diene-3,5-dione), a polyphenol found in the natural product turmeric, specifically inhibits p300-HAT activity [7]. They found that curcumin inhibited the phenylephrine (PE)-stimulated acetylation of GATA4 and histones and suppressed cardiomyocyte hypertrophy and the transcriptional activity of cardiac hypertrophic response factors. Subsequent studies have shown that oral administration of curcumin 50 mg/kg inhibits the progression of heart failure in rat models of chronic heart failure, with causes hypertension and myocardial infarction [8,9]. This body of research suggests that the inhibition of p300-HAT activity in cardiomyocytes prevents heart failure.

In this study, we screened a library of natural extracts to identify compounds that suppress cardiomyocyte hypertrophy, a risk factor for heart failure. Among several potential compounds, we focused on anserine (Figure 1), a histidine-containing dipeptide (HCD) that is abundant in the skeletal and cardiac muscle, kidneys and brains of fish, birds and mammals.

Other prominent human HCDs include carnosine (β-alanyl-L-histidine), of which anserine (β-alanyl-N-methylhistidine) is a methylated analog [10]. The known physiological roles of anserine and carnosine include increased exercise tolerance (anti-fatigue) due to pH buffering [11,12], antioxidant effects [13], inhibition of protein glycation [14] and aldehyde detoxification [15].

The cardiovascular effects of HCDs have recently been studied using HCD carnosine, which has been reported to increase contractility in isolated hearts [16] and to protect cardiomyocytes after ischemia [17]. In addition, Gonçalves et al. found impaired cardiac contraction, cardiac dilation and abnormal ECG findings in carnosine synthase-knockout rats. The absence of HCDs in these knockout rats reduced the Ca^2+^ removal rate within the myocardium, resulting in impaired relaxation and abnormal cardiac function [18]. Notably, anserine is not as abundant in the heart as carnosine [17], and its direct effects on cardiac function have not yet been elucidated. In this study, we investigated the effects of anserine on heart failure using primary cultured cardiomyocytes and a mouse model of heart failure.

## 2. Results

### 2.1. Anserine Suppressed Phenylephrine (PE)-Induced Cardiomyocyte Hypertrophic Responses

To investigate whether anserine suppresses PE-induced cardiomyocyte hypertrophy, primary cultured cardiomyocytes prepared from neonatal rats were treated with 1, 3 or 10 mM anserine for 2 h and then stimulated with 30 µM PE for 48 h. Immunofluorescence staining showed that anserine significantly suppressed PE-induced cardiomyocyte hypertrophy (Figure 2A,B). Quantitative RT-PCR revealed that anserine significantly and dose-dependently inhibited the PE-induced increases in the mRNA levels of the hypertrophy-related genes ANF and BNP (Figure 2C,D). These results suggest that anserine inhibits PE-induced hypertrophic responses in cultured cardiomyocytes.

### 2.2. Anserine Suppressed p300-HAT Activity in Cultured Cardiomyocytes and In Vitro

To determine whether anserine inhibits p300-HAT activity, histone fractions were prepared from cultured cardiomyocytes. Western blotting indicated that anserine inhibited the PE-induced acetylation of histone H3K9 in cultured cardiomyocytes (Figure 3A,B).

To confirm whether anserine directly suppresses p300-HAT activity, an in vitro p300-HAT assay using a recombinant p300-HAT domain (residues 1284–1674) was performed. Histone H3K9 acetylation was significantly inhibited by anserine (Figure 3C). The IC_50_ value of anserine was calculated using the sigmoid dose–response curves with variable slopes, resulting in an estimated value of 1.87 mM (Figure 3D). This suggests that anserine directly inhibits p300-HAT activity and suppresses PE-induced hypertrophic responses in cardiomyocytes, at least in part by inhibiting p300-HAT activity.

### 2.3. Marine Active^®^ Suppressed PE-Induced Cardiomyocyte Hypertrophy

Marine Active^®^ contains a high concentration of anserine (38% anserine) and other molecules (L-histidine: 26.2%, carnosine: 3.4%, others: 32.4%). To investigate whether Marine Active^®^ suppresses PE-induced cardiomyocyte hypertrophy, primary cultured cardiomyocytes were treated with 10 mM anserine, 10.6 mM L-histidine, 0.9 mM carnosine or 6.3 mg/mL Marine Active^®^ (containing 10 mM anserine, 10.6 mM L-histidine and 0.9 mM carnosine) for 2 h and then stimulated with 30 µM PE for 48 h. Immunofluorescence staining showed that 6.3 mg/mL Marine Active^®^ suppressed PE-induced cardiomyocyte hypertrophy significantly more than 10 mM anserine (Figure 4A). Neither 10.6 mM L-histidine nor 0.9 mM carnosine inhibited PE-stimulated cardiomyocyte hypertrophy (Figure 4B).

We next performed an in vitro p300-HAT assay using 6.3 mg/mL Marine Active^®^, 10.6 mM L-histidine and 1.0 mM carnosine. The acetylation of histone H3K9 was significantly inhibited by 6.3 mg/mL Marine Active^®^ but not by 10.6 mM L-histidine or 0.9 mM carnosine (Figure 4C). The IC_50_ value of the p300-HAT inhibitory effect of Marine Active^®^ was calculated using the sigmoid dose–response curves with variable slopes, resulting in an estimated value of 4.64 mg/mL (2.70 mM anserine) (Figure 4D). These results suggest that Marine Active^®^ inhibits p300-HAT activity and cardiomyocyte hypertrophy to the same extent as anserine.

### 2.4. Marine Active^®^ Suppressed Transverse Aortic Constriction (TAC)-Induced Systolic Dysfunction

The TAC mouse model was then used to investigate whether anserine suppresses the development of heart failure in vivo. One day after TAC surgery, the mice were allocated to one of three groups: Marine Active^®^ low dose (anserine: 60 mg/kg/day), Marine Active^®^ high dose (anserine: 200 mg/kg/day) or vehicle (0.5% CMC-Na in saline) as a control. Daily oral administration was continued for 8 weeks. Representative echocardiographic images are displayed in Figure 5A, and the echocardiographic and hemodynamic parameters are presented in Table 1. Eight weeks after treatment, the vehicle group demonstrated significantly decreased left ventricular fractional shortening (FS) and increased posterior wall thickness (PWd) as well as interventricular septum thickness in diastole (IVSD). These changes were suppressed by a high dose of Marine Active^®^ treatment. Pressure overload induced cardiac hypertrophy (Figure 5B) and also displayed significant increases in heart weight vs. body weight (HW/BW, Figure 5C) and heart weight vs. tibia length (HW/TL, Figure 5D) ratios in the vehicle group. These increases were also attenuated by a high dose of Marine Active^®^ treatment. There were no differences in liver weight vs. BW and TL or in kidney weight vs. BW and TL among each group (Appendix A). These results indicate that a high dose of Marine Active^®^ significantly suppresses cardiac hypertrophy and prevents systolic dysfunction in the TAC mouse model.

### 2.5. Marine Active^®^ Suppressed TAC-Induced Cardiac Hypertrophy

To investigate whether Marine Active^®^ suppresses TAC-induced cardiac hypertrophy in mice, hematoxylin and eosin (HE) staining was performed (Figure 6A). A high dose of Marine Active^®^ treatment significantly, but not completely, suppressed TAC-induced cardiomyocyte hypertrophy (Figure 6B). In addition, the liver and kidney tissues were stained with HE staining. The abnormalities of these tissues were not observed (Appendix A). Next, to investigate whether Marine Active^®^ suppresses TAC-induced hypertrophy-related gene transcription, quantitative RT-PCR was performed. A high dose of Marine Active^®^ treatment significantly suppressed the transcription of ANF and BNP (Figure 6C,D). These findings suggest that a high dose of Marine Active^®^ inhibits the TAC-induced cardiac hypertrophic response in mice.

### 2.6. Marine Active^®^ Suppressed TAC-Induced Cardiac Fibrosis

To investigate whether Marine Active^®^ suppresses TAC-induced cardiac fibrosis, Masson’s trichrome (MT) staining was performed (Figure 7A). Marine Active^®^ treatment significantly, but not completely, inhibited TAC-induced perivascular fibrosis (Figure 7B). Next, to investigate whether Marine Active^®^ suppresses TAC-induced fibrotic gene transcription in mice, quantitative RT-PCR was performed. A high dose of Marine Active^®^ significantly suppressed the TAC-induced upregulation of the mRNA levels of fibrosis-related genes, collagen 1A1 and collagen 3A1 (Figure 7C,D). These results indicate that a high dose of Marine Active^®^ suppresses TAC-induced cardiac fibrosis in mice.

### 2.7. Marine Active^®^ Suppressed TAC-Induced Acetylation of Histone H3K9

To investigate whether Marine Active^®^ suppresses the TAC-induced acetylation of histone H3K9 in mice, Western blotting was performed using histone fractions from mouse hearts. The results indicated that a high dose of Marine Active^®^ treatment significantly inhibited the TAC-induced acetylation of histone H3K9 (Figure 8A,B).

### 2.8. Oral Administration of Marine Active^®^ Increased the Content of Anserine in Heart Tissue

To clarify whether the administration of Marine Active^®^ accumulated anserine in the heart, we measured cardiac anserine content using the heart tissue of each group. Chronic pressure overload reduced the anserine content in heart tissue compared to sham surgery. A high dose of Marine Active^®^ significantly increased cardiac anserine content compared to the TAC group with vehicle treatment. The amount of anserine in low doses of the Marine Active^®^ group rarely increased. This finding supported that the high dose, and not a low dose, of the Marine Active group exerted a therapeutic effect of anserine on the development of heart failure in vivo.

## 3. Discussion

Chronic heart failure is exacerbated by hypertension and other stressors that cause cardiac hypertrophy and fibrosis, such as cardiac remodeling [2,3]. In this study, the HCD anserine suppressed both the PE-induced hypertrophy and the increases in ANF and BNP mRNA levels in primary cultured cardiomyocytes in a dose-dependent manner. Furthermore, Marine Active^®^ containing anserine prevented the impairment of left ventricular systolic function and inhibited not only the hypertrophy of individual cardiomyocytes but also perivascular fibrosis in mice models of pressure overload induced through TAC surgery.

Anserine inhibited the acetylation of histone H3K9 in primary cultured cardiomyocytes and in vivo, suggesting that it directly inhibits p300-HAT activity. Anserine is a β-alanyl-N-methylhistidine, a methylated analog of the histidine residue of carnosine. As Marine Active^®^ also contains histidine and carnosine in addition to anserine, we thought that these compounds in Marine Active^®^ may have inhibited p300-HAT activity in a way similar to that of anserine. However, our follow-up experiment revealed that anserine and not histidine nor carnosine inhibit p300-mediated H3K9 acetylation in the same concentrations as Marine Active (Figure 3D). Our findings suggest that anserine in Marine Active^®^ is exerted primarily as a p300-HAT inhibitor and exhibits anti-hypertrophic effects in cardiomyocytes.

Curcumin, anacardic acid, L002, C646, metformin, EPA and DHA inhibit the HAT activity of p300 and prevent the progression of heart failure [8,19,20,21,22]. Anserine is advantageous over curcumin in that it is more water-soluble. However, the disadvantage is that the concentration at which anserine inhibited primary cultured cardiomyocyte hypertrophy, 10 mM, is 1000-fold higher than the concentration at which curcumin does the same, 10 µM. This is probably because highly liposoluble substances are easily transported into cardiomyocytes, as the heart derives most of the high energy required for contraction from the mitochondrial oxidation of fatty acids [23,24]. In contrast, the required in vivo anserine dose of 200 mg/kg is 4-fold higher than the required curcumin dose of 50 mg/kg. Anserine and carnosine are incorporated into the cell via intestinal peptide transporters such as PEPT1 and PEPT2 [25]. This difference in concentration may be related to the different metabolic pathways of curcumin and anserine.

Carnosine is the primary HCD in humans. Unlike rodents, humans have high serum carnosinase activity, resulting in poor stability of carnosine in serum and very low bioavailability [26]. Because HCDs are present in meat and fish, the daily consumption of these dipeptides in an omnivorous diet also affects the carnosine content of the muscle. The daily intake of anserine and carnosine in Japanese elderly people is approximately 579 mg/day (65% anserine, 35% carnosine) [27]. Anserine has a 5- to 10-fold longer half-life than carnosine in human serum [28,29]. Yeum et al. examined the plasma HCD concentration in humans after ingestion of pure carnosine (450 mg) or dietary HCDs, beef (anserine: 43 mg, carnosine: 343 mg), chicken (anserine: 660 mg, carnosine: 322 mg) and chicken broth (anserine 475 mg, carnosine 316 mg). Although plasma carnosine concentration was below the detection limit for all forms, plasma anserine concentrations were significantly increasing after the ingestion of chicken and chicken broth (*C*_max_: 2.7 µM and 0.78 µM, respectively) [29]. Everaert et al. performed similar experiments with pure anserine (4 mg/kg). They found *C*_max_ values of 0.5 µmol/L for anserine and less than the detection limit for carnosine [28]. These findings indicate that anserine is more resistant than carnosine to hydrolysis by serum carnosinase. Recently, carnosinase inhibitors have been reported to increase HCD levels in the body [30]. In other words, the simultaneous administration of anserine and a carnosinase inhibitor can be expected to maintain blood levels of anserine, allowing a greater ameliorative effect of anserine on heart failure.

Our collaborators showed that, when humans were orally administered either an-serine alone (2.0 g/60 kg) or Marine Active^®^ (19.4 g/60 kg, providing anserine at 2.0 g/60 kg) dissolved in water, there were no significant differences in the kinetic parameters *AUC*_0-4_, tmax, *t*_1/2_ or *C*_max_ [31]. This indicates that the intestinal absorption and blood clearance of anserine are largely unaffected by other factors such as carnosine, other amino acids, proteins and dextrin [31]. In this study, we detected the anserine content in heart tissue and found that a high dose of Marine Active^®^ significantly increased the cardiac level of anserine (Figure 8C). Thus, as Marine Active^®^ has the same effect as anserine alone, we speculate that cardiac accumulated anserine is the main factor in the inhibitory effect of Marine Active^®^ on cardiac hypertrophy and cardiac dysfunction.

Interestingly, chronic pressure overload, such as TAC surgery, induced the reduction of cardiac anserine levels. Anserine is synthesized from carnosine by carnosine-N-methyltransferase enzyme (CARNMT1) and produced by transferring a methyl group onto carnosine [32]. Drozak et al. showed that mouse, human and chicken carnosine synthase 1 (CARNS1) can synthesize anserine from β-alanine and 1-methyl-histidine [33]. Carnosine dipeptidase 1 and 2 (CNDP1 and CNDP2) can degrade both carnosine and anserine [34]. Recently, anserine is present in the human cardiac muscle [35]. Anserine has been poorly studied compared to its analog, carnosine and its biological relevance, especially in the heart, is largely unknown. Our recent study showed anserine’s protective effects on cardiac hypertrophy and systolic dysfunction. Since the abnormality of anserine metabolism may be involved in the development of cardiac hypertrophy and heart failure, it requires further analysis in the future.

In this study, Marine Active^®^ inhibited hypertrophy in primary cultured cardiomyocytes significantly more than anserine alone (Figure 3), suggesting that components other than anserine may affect anti-hypertrophic effects. Similarly, in the in vitro HAT assay, the IC_50_ of Marine Active^®^ was 2.70 mM in terms of anserine, while that of an-serine alone was 1.87 mM. There was almost no difference in IC_50_ between anserine and Marine Active^®^. Marine Active^®^ is made from skipjack and yellowfin tuna extracts and contains 38% anserine, 26.2% L-histidine, 3.4% carnosine, 7% other amino acids and 18.4% protein. The equal amounts of L-histidine alone and carnosine alone did not inhibit the HAT activity of p300 in vitro. Some intercellular peptidases, such as CNDP2 hydrolyze anserine, carnosine and dipeptides, contain histidine [34]. It is possible that the carnosine and histidine in Marine Active^®^ competitively affect CNDP2 activity and attenuate the degradation of anserine, resulting in an increased hypertrophic effect of Marine Active^®^ over anserine. The structure–activity relationship analysis of anserine for HAT-inhibitory and anti-hypertrophic effects is necessary to understand the molecular mechanism of anserine for future studies.

## 4. Materials and Methods

### 4.1. Materials

Anserine (Figure 1), carnosine, L-histidine, 1-methyl-histidine, β-alanine, phenylephrine, acetonitrile, trifluoroacetic acid, trichloroacetic acid, aminoadipic acid, triethylamine and phenyl isothiocyanate were purchased from Fujifilm Wako Pure Chemical (Osaka, Japan). Hydrophilic nylon syringe filters (0.45 μm) were purchased from Hamach Scientific (Xi’an, China). Anserine, carnosine, L-histidine, 1-methyl-histidine and β-alanine were dissolved in distilled water and filtered before used. Marine Active^®^ was provided by Yaizu Suisankagaku Industry (Shizuoka, Japan) and stored at 4 °C. Marine Active^®^ contains 38% anserine, 26.2% L-histidine, 3.4% carnosine and 32.4% others.

### 4.2. Animals

Neonatal Sprague Dawley (SD) rats were purchased from Japan SLC (Shizuoka, Japan). Seven- to eight-week-old C57BL/6j male mice were obtained from Japan CLEA Japan (Tokyo, Japan). This study was approved by the Ethics Committee of the University of Shizuoka (US176278 for primary cultured cardiomyocytes, US176278, US176279 for animal experiments) and Kyoto Medical Center (KMC3-29-1 for animal experiments). All procedures were conducted following the US National Institutes of Health (NIH) Guide for the Care and Use of Laboratory Animals.

### 4.3. Neonatal Rat Cardiomyocytes

Neonatal rat cardiomyocytes were isolated from the ventricles of 1 to 3-day-old Sprague Dawley rats, plated on indicated dishes in DMEM containing 10% fetal bovine serum and cultured as described previously [8,36]. The plated cells were pre-treated with 1, 3 or 10 mM of anserine for 2 h and then stimulated with 30 μM of PE for 48 h.

### 4.4. Fluorescence Staining and Confocal Microscopy

Immunofluorescence staining and confocal microscopy were performed as described [20]. In brief, cultured cardiomyocytes stimulated with PE for 48 h were stained. The primary antibody was the mouse anti-α-actinin antibody (Sigma-Aldrich, Saint Louis, MO, USA), and the secondary antibody was Alexa Fluor 555-conjugated anti-mouse IgG (Invitrogen, Waltham, MA, USA). Hoechst33258 (Dojinjo, Kumamoto, Japan) was used for staining nuclei. ArrayScan^TM^ (Thermo Fisher Scientific, Carlsbad, CA, USA) automatically measured three hundred α-actinin-positive cardiomyocyte surface areas.

### 4.5. RNA Isolation and Quantitative Reverse Transcription PCR

Total RNA was extracted from cardiomyocytes and mouse ventricular myocardium using the TRI Reagent^®^ (Nacalai Tesque, Kyoto, Japan) according to the manufacturer’s instructions. Quantitative reverse transcription PCR (RT-PCR) was performed as previously described [8,36]. cDNA was generated via ReverTra Ace^®^ qPCR RT Master Mix (TOYOBO, Osaka, Japan). Quantitative RT-PCR analysis was conducted with KOD SYBR^®^qPCR Mix (TOYOBO). All qPCR reactions were performed under the following conditions: 98 °C for 2 min, 98 °C for 10 s, followed by 40 cycles of amplification at 50 °C for 10 s and 68 °C for 30 s. All reactions for the targets (ANF, BNP, collagen 1 alpha 1 and collagen 3 alpha 1) and reference (18S) genes were completed in duplicate, and the average value was used for subsequent quantification. The relative quantities of complementary DNA were calculated from cycle thresholds and were normalized to the *18S* gene using the 2−∆∆Ct method. The primer sequences used in this study are listed in Table 2. The ANF, BNP and 18S primers have sequence homologies between mice and rats.

### 4.6. Western Blotting

Histone fractions were extracted from cardiomyocytes or the mice’s left ventricle (LV), and then Western blotting was performed [8,37]. Each loading sample containing the same amount (15 µg/10 µL) of protein was transferred to a Nitrocellulose membrane (Hybond ECL, GE healthcare, Chicago, IL, USA). The membrane was incubated with specific antibodies for anti-acetyl-histone H3K9 (#9649, Cell Signaling Technology, Danvers, MA, USA) and anti-histone H3 (#4499, Cell Signaling Technology) antibodies. Western blotting signals were visualized with an Amersham Imager 680 imager (GE Healthcare, Chicago, IL, USA) and quantified with NIH ImageJ software (version 1.52a).

### 4.7. In Vitro p300-HAT Assay

In vitro p300-HAT assays were performed as described previously [38]. Briefly, a purified p300-HAT recombinant domain (residues 1284–1674) and 5 μg of core histones from calf thymus (Worthington, Columbus, OH, USA) were incubated with or without anserine at room temperature for 30 min. Adding acetyl-CoA to each sample initiated the reactions, which were then incubated for 30 min. All samples were subjected to SDS-PAGE, and Western blotting was performed using anti-acetyl-histone H3K9 and anti-histone H3 antibodies. The 50% inhibitory concentration (IC_50_) was calculated from the created concentration–response curves.

### 4.8. TAC Surgery

Male C57BL/6J mice (7 to 8 weeks of age) were subjected to TAC or sham operation. In brief, mice were anesthetized with 1.0–1.5% isoflurane. Aortic constriction was performed by ligating the transverse thoracic aorta between the innominate artery and left common carotid artery with a 27-gauge needle using a 5-0 nylon string as previously described [39]. The pressure gradient in the thoracic aorta was measured via Doppler echocardiography to confirm significant pressure overload in all mice. The sham operation was identical, except that the thread was not ligated.

### 4.9. Drug Treatments

The 21 mice received daily oral treatment of vehicle (0.5% CMC-Na in saline, *n* = 7), low-dose Marine Active^®^ (60 mg/kg of anserine, *n* = 7) or high-dose Marine Active^®^ (200 mg/kg of anserine, *n* = 7) from one day after TAC surgery. The compounds were dissolved with 0.5% CMC-Na in saline and administrated to the mice orally via gastric gavage once a day for 8 weeks. The sham mice (*n* = 7) were treated with the vehicle.

### 4.10. Echocardiography

The cardiac function of the mice was evaluated with two-dimensional (M-mode) echocardiography using a 10–12 MHz probe (Sonos 5500 Ultrasound System; Philips, Amsterdam, The Netherlands) as described previously [39]. Specifically, the mice were subjected to 1.0–1.5% isoflurane anesthesia and positioned supine on an operating table constantly maintained at 37 °C. Various parameters, including IVSd, PWd, left ventricular internal diameter end-diastole (LVIDd) and left ventricular internal diameter end-systole (LVIDs) were obtained by analyzing the original M-mode tracings over the three cardiac cycles. The calculation used to determine FS was (LVIDd − LVIDs)/LVIDd × 100 (%). Ejection fraction (EF) was calculated as LV end-diastolic volume (LVEDV) − LV end-systolic volume (LVESV)/LVEDV × 100 (%). To eliminate bias, image analysis was performed by a researcher who was blinded to the group assignment.

### 4.11. Histological Analysis

Eight weeks after sham or TAC surgery, mice were euthanized to assess the extent of cardiac hypertrophy and fibrosis [40]. To estimate the cardiac hypertrophy, HW/TL and HW/BW ratios were calculated. The excised hearts were fixed in 4% paraformaldehyde solution for 24 h and embedded in paraffin following standard histological protocols. Heart sections (5 μm) were subjected to HE staining for the cross-sectional areas of the cardiomyocytes and MT staining to assess collagen deposition. At least 50 cells per slide were analyzed to determine the cell area. Perivascular fibrosis was quantified by measuring the area that was stained positive with Masson’s trichrome staining using ImageJ and was expressed as a percentage of the intramyocardial coronary artery, being greater than 50 μm in each mouse.

### 4.12. Measurement of Anserine Content in Heart Tissue

Anserine content was measured in heart tissue according to the method of Shigemura et al. [41] with some modifications. In brief, heart tissue (100 mg) was added in phosphate-buffered saline and homogenized using Micro Smash^TM^ MS-100R at 5000 rpm, at 4 °C and for 80 s. After centrifugation at 12,000× *g*, at 4 °C and for 10 min, the supernatant was added trichloroacetic acid (5%), vortexed, centrifugated at 12,000× *g*, at 4 °C and for 10 min and filtrated through a 0.45 μm hydrophilic nylon membrane. Aminoadipic acid was used as an internal control. Twenty μL of this solution was lyophilized. After phenylthiocarbamylation, the sample was dissolved in solvent A (0.01% trifluoroacetic acid pH 3.0) and solvent B (60% acetonitrile) (5:15, *v*/*v*), filtered and analyzed using an HPLC system and Wakopak Wakosil-PTC (4.0 × 250 mm) column (Fujifilm Wako Pure Chemical, Osaka, Japan) at a flow rate of 0.5 mL/min. The column was equilibrated with 15% solvent B. The gradient profile was as follows: 0–30 min, 15–75% B; 30–35 min, 75–100 B; 35–40 min, 100% B; 40–50 min, 15% B. The column was maintained at 45 °C. The absorbance at 254 nm was monitored. The anserine content was calculated using anserine standards, and a calibration curve was constructed over a concentration range of 0.1 nM to 10 nM.

### 4.13. Statistics

Results are presented as mean ± SD. Statistical comparisons were performed using one-way ANOVA with the Tukey–Kramer test for post hoc multiple comparisons (Stat View 5.0 software, SAS Institute Inc, Cary, NC, USA). A *p*-value of <0.05 was considered statistically significant.

## 5. Conclusions

This study demonstrates that anserine attenuates the development of cardiac hypertrophy and heart failure via p300-HAT inhibition, both in primary cultured cardiomyocytes and in a mouse model of heart failure. Because aging induces the expression of the carnosinase gene and decreases serum HCD concentrations [42,43], and because anserine is more resistant to degradation than carnosine in humans [28,44], anserine may be clinically applicable in the prevention of heart failure.

## Figures and Tables

**Figure 1 ijms-25-02344-f001:**
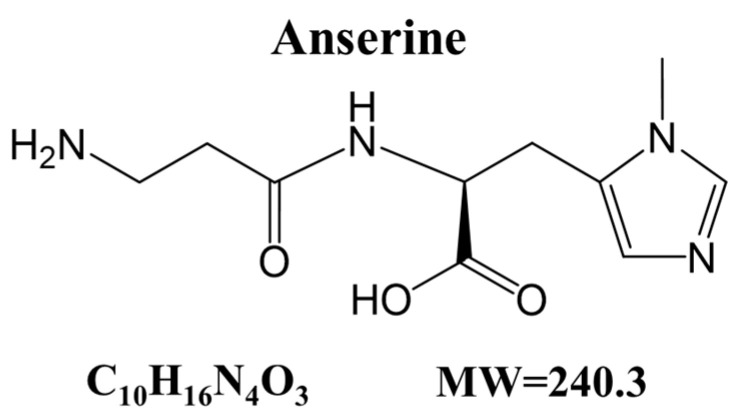
Chemical structure of anserine.

**Figure 2 ijms-25-02344-f002:**
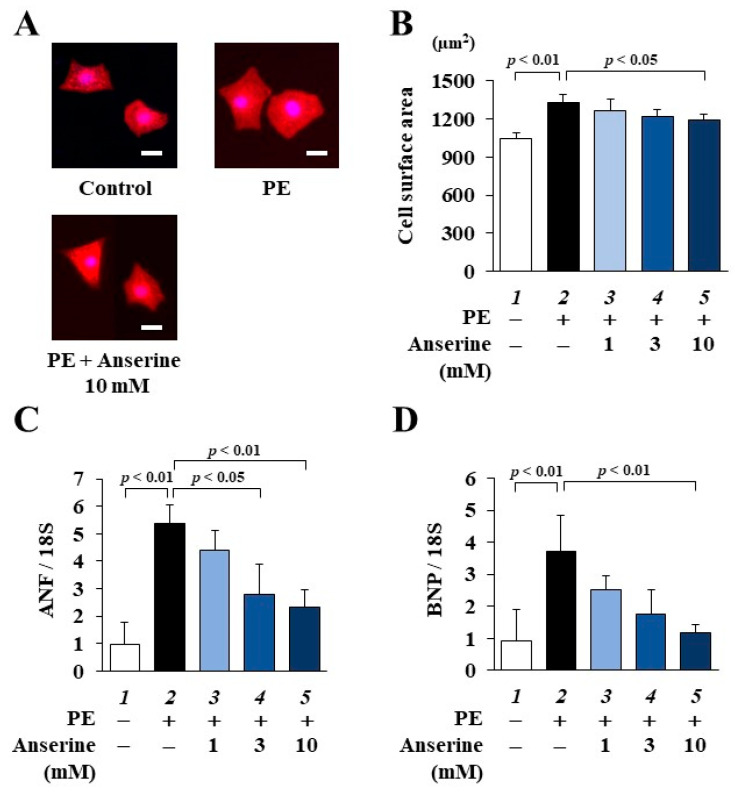
Phenylephrine-induced hypertrophic responses were inhibited by anserine in cultured cardiomyocytes. Effects of anserine on primary cultured cardiomyocytes. (**A**) Representative images of immunofluorescence staining for anti-α-actinin (red) and Hoechst 33258 (blue) in primary cultured cardiomyocytes. Scale bar = 20 μm. (**B**) The results of cardiomyocyte surface area. All data are presented as mean ± SD (*n* = 5). (**C**,**D**) Quantitative RT-PCR analysis showed the mRNA levels of ANF (**C**) and BNP (**D**). Data are presented as mean ± SD (*n* = 3).

**Figure 3 ijms-25-02344-f003:**
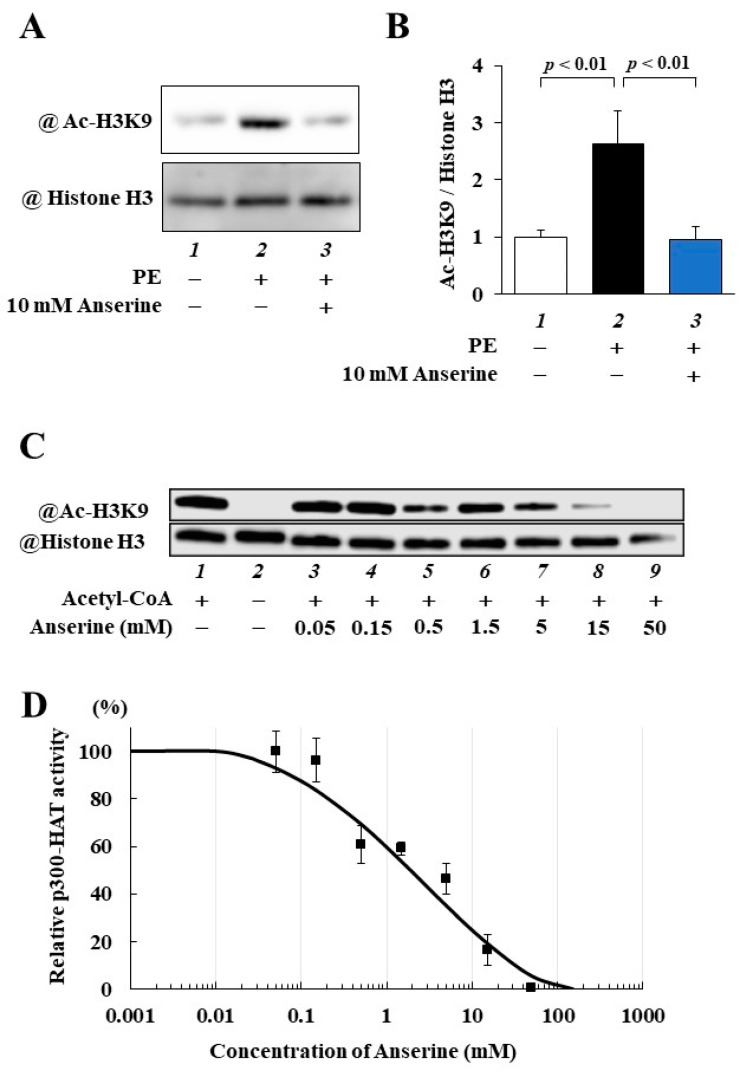
p300-HAT activity was inhibited by anserine. (**A**,**B**) Effects of anserine on primary cultured cardiomyocytes. (**A**) Representative photos of Western blotting in primary cultured cardiomyocytes. (**B**) Acetylated histone H3K9/total histone H3. Data are presented as mean ± SD (*n* = 3). (**C**,**D**) The results of in vitro p300-HAT assay. (**C**) Representative photos of Western blotting in vitro. (**D**) Concentration–response curve of acetyl-histone H3K9/histone H3 vs. common logarithm (concentrations). The IC_50_ value for the inhibition of p300 HAT activity with anserine was 1.87 mM (*n* = 3).

**Figure 4 ijms-25-02344-f004:**
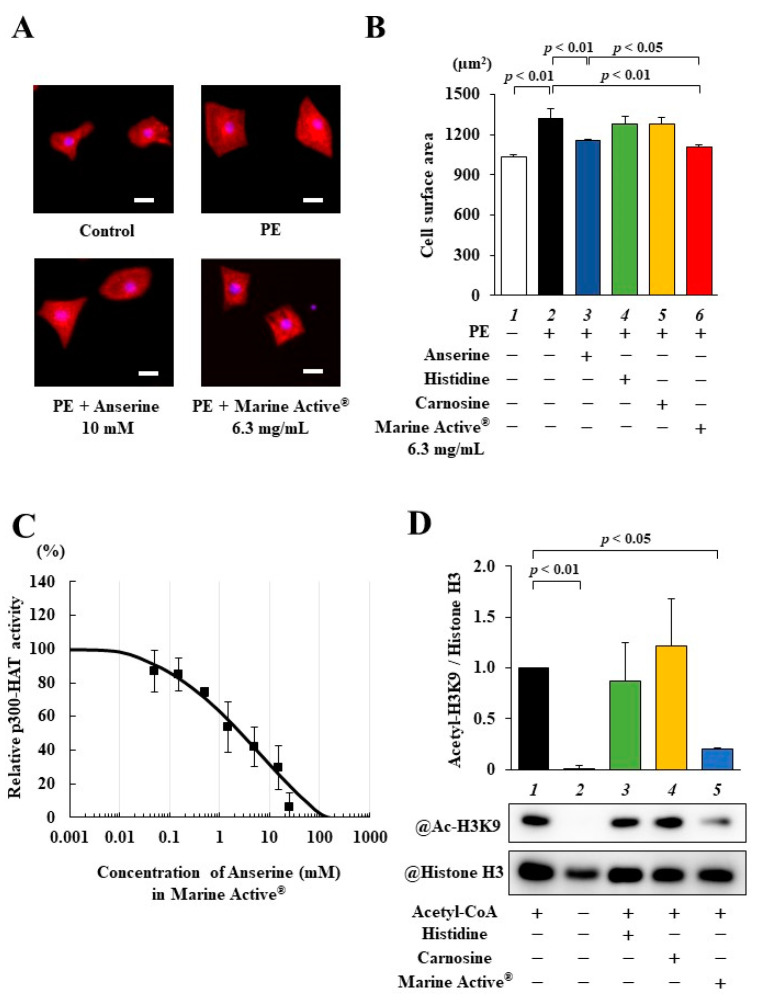
Phenylephrine-induced cardiomyocyte hypertrophy was inhibited by Marine Active^®^ in cultured cardiomyocytes. (**A**,**B**) Effects of Marine Active^®^ on primary cultured cardiomyocytes. (**A**) Representative images of immunofluorescence staining for anti-α-actinin (red) and Hoechst 33258 (blue) in primary cultured cardiomyocytes. Scale bar: 20 μm. (**B**) The results of cardiomyocyte surface area. All data are presented as mean ± SD (*n* = 4). (**C**,**D**) The results of in vitro p300-HAT assay using 10 mM anserine, 10.6 mM L-histidine, 1.0 mM carnosine and 6.3 mg/mL Marine Active^®^ (10 mM anserine, 10.6 mM L-histidine and 1.0 mM carnosine). (**C**) Representative photos of Western blotting in vitro. Western blotting was performed with anti-acetyl-histone H3K9 and anti-histone H3 antibodies. (**D**) Concentration–response curve of acetyl-histone H3K9/histone H3 vs. common logarithm (concentrations). The IC_50_ value for the inhibition of p300 HAT activity with Marine Active^®^ was 4.64 mg/mL (2.70 mM anserine) (*n* = 3).

**Figure 5 ijms-25-02344-f005:**
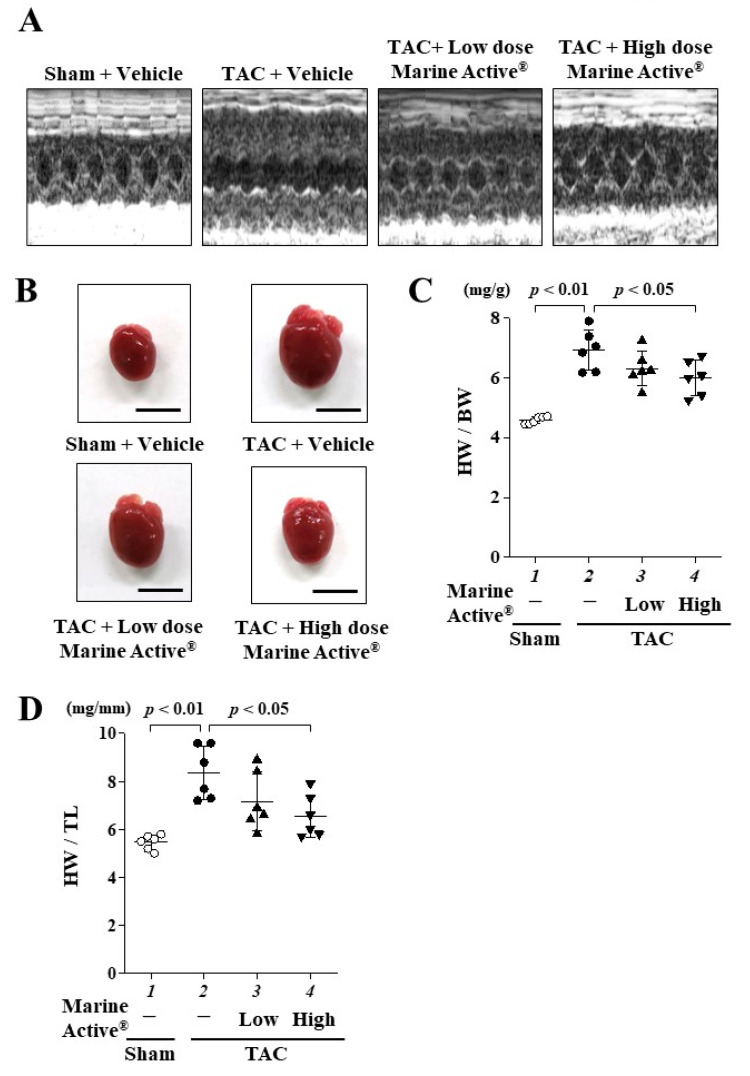
TAC-induced systolic dysfunction was suppressed by Marine Active^®^. The results of echocardiography 8 weeks after treatment. (**A**) Representative M-mode echocardiograms of mouse heart. (**B**) Representative images of whole hearts from the mice. Scale bar: 5 mm. (**C**) The results of HW/BW. Data are presented as mean ± SD (*n* = 6). (**D**) The results of HW/TL. Data are presented as mean ± SD (*n* = 6).

**Figure 6 ijms-25-02344-f006:**
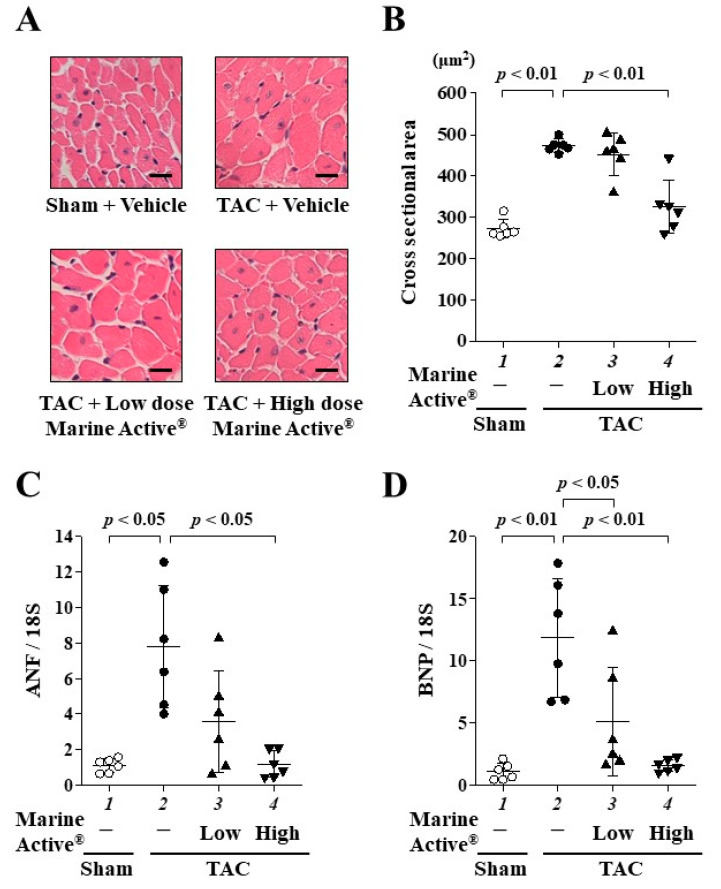
TAC-induced cardiac hypertrophy was inhibited with Marine Active^®^. (**A**) Representative images of HE-stained sections of the mice hearts. Magnification: ×400. Scale bar: 20 μm. (**B**) The results of myocardial cell diameter. Data are presented as mean ± SD (*n* = 6). (**C**,**D**) Quantitative RT-PCR analysis showed the mRNA levels of ANF (**C**) and BNP (**D**). Data are presented as mean ± SD (*n* = 6).

**Figure 7 ijms-25-02344-f007:**
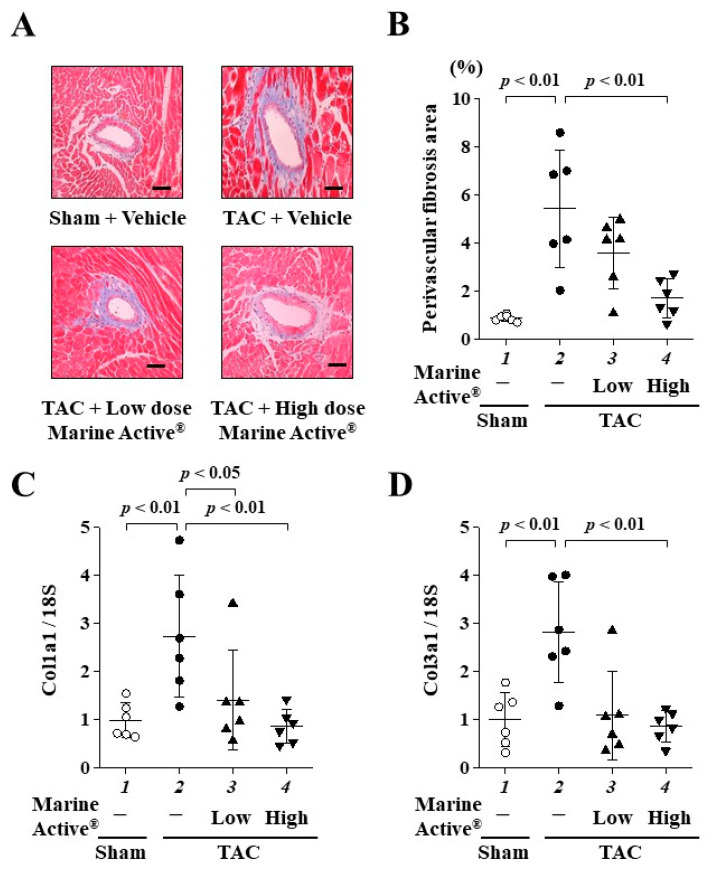
TAC-induced cardiac fibrosis was inhibited by Marine Active^®^. (**A**) Representative images of MT-stained perivascular fibrosis area of LV myocardium from the mice. Magnification: ×400. Scale bar: 50 μm. (**B**) The results of perivascular fibrosis in the LV. Data are presented as mean ± SD (*n* = 6). (**C**,**D**) Quantitative RT-PCR data for (**C**) Col 1A1, (**D**) Col 3A1 and 18S. Data are presented as mean ± SD (*n* = 6).

**Figure 8 ijms-25-02344-f008:**
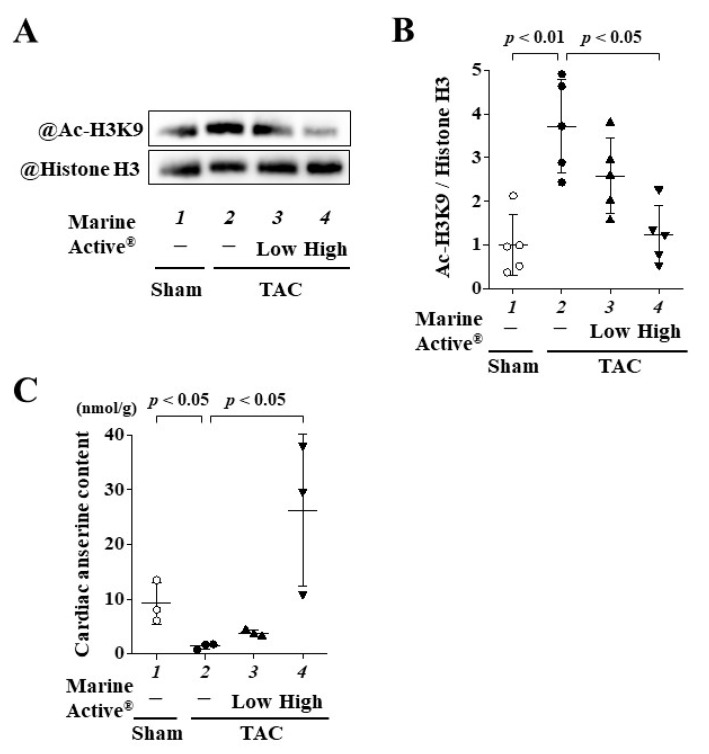
TAC-induced acetylation of histone H3K9 was inhibited with Marine Active^®^. (**A**) Representative photos of Western blotting in vivo. Western blotting was performed with anti-acetyl-histone H3K9 and anti-histone H3 antibodies. (**B**) Acetylated histone H3K9/total histone H3. Data are presented as mean ± SD (*n* = 4). (**C**) The contents of anserine in heart tissue were determined by HPLC-UV. Data are presented as mean ± SD (*n* = 3).

**Table 1 ijms-25-02344-t001:** Echocardiographic parameters at 8 weeks after TAC operation.

	Sham	TAC		
	Vehicle	Vehicle	Low Dose of Marine Active^®^ (Anserine 60 mg/kg)	High Dose of Marine Active^®^ (Anserine 200 mg/kg)
LVIDs (mm)	1.33 ± 0.12	1.52 ± 0.17	1.39 ± 0.13	1.15 ± 0.09 ^##^
LVIDd (mm)	2.50 ± 0.17	2.35 ± 0.21	2.24 ± 0.17	2.18 ± 0.21 *
PWd (mm)	1.19 ± 0.10	1.77 ± 0.11 **	1.72 ± 0.14 **	1.38 ± 0.12 *^##^
IVSd (mm)	1.21 ± 0.10	1.66 ± 0.15 **	1.63 ± 0.13 **	1.39 ± 0.10 *^##^
FS (%)	46.6 ± 3.6	35.5 ± 3.2 **	38.1 ± 1.5 **	47.0 ± 2.2 ^##^
EF (%)	83.9 ± 3.5	72.2 ± 3.7 **	75.4 ± 2.1 **	84.4 ± 2.4 ^##^
HR (bpm)	573 ± 20	607 ± 17	620 ± 12	626 ± 10
BW (g)	27.0 ± 1.6	27.3 ± 1.9	26.8 ± 2.1	26.3 ± 1.4

The data are shown as mean ± SD. *: *p* < 0.05, **: *p* < 0.01. Versus sham + vehicle, ^##^: *p* < 0.01 versus TAC + vehicle. LVIDs: left ventricular internal diameter end-systole, LVIDd: left ventricular internal diameter end-diastole, PWd: posterior LV wall thickness in diastole, IVSd: Interventricular septum thickness in diastole, FS: fractional shortening, EF: ejection fraction, HR: heart rate, BW: body weight.

**Table 2 ijms-25-02344-t002:** The primer sequences for quantified RT-PCR.

Target Gene	Forward	Reverse
*Nppa* (ANF)	5′-AGGCCATATTGGAGCAAATC-3′	5′-CATCTTCTCCTCCAGGTGGT-3′
*Nppb* (BNP)	5′-GATTCTGCTCCTGCTTTTCC-3′	5′-CATCGTGGATTGTTCTGGAG-3′
*Col1a1* (Collagen alpha 1)	5′-AAGAAGACATCCCTGAAGTCA-3′	5′-TTGTGGCAGATACAGATCAAG-3′
*Col3a1* (Collagen 3 alpha 1)	5′-CCCAACCCAGAGATCCCATT-3′	5′-GAAGCACAGGAGCAGGTGTAGA-3′
*RNA18S1* (18S)	5′-CTTAGAGGGACAAGTGGCG-3′	5′-GGACATCTAAGGGCATCACA-3′

## Data Availability

The data presented in this study are available upon request from the corresponding author.

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
