# Peer review of "Anserine, a Histidine-Containing Dipeptide, Suppresses Pressure Overload-Induced Systolic Dysfunction by Inhibiting Histone Acetyltransferase Activity of p300 in Mice"

_ijms, 2024, doi:10.3390/ijms25042344_

Round 1
Reviewer 1 Report
Comments and Suggestions for Authors
tThe authors discussed a hot research area which is the treatment of cardiac hypertrophy. they suggested that Anserine would be a potential treatment for pressure overload-induced cardiac hypertrophy.
· The authors nicely presented their data and showed that Anserine prevented PE-induced hypertrophy in vitro and Marine active which contains Anserine as a main component reduced the effect of TAC-induced hypertrophy. but the results did not indicate whether is it the effect of Anserine alone or the combination of the 3 compounds. the authors tried to discuss this issue but here are some ideas that might help answer this question
1) test the effect of different concentrations of carnosine, and histidine on PE-induced hypertrophy in vitro alone and a combination of the most effective doses.
2) Test the effect of Anserine alone in vivo instead of Marine.
· The authors showed that Marine protected the heart against TAC-induced hypertrophy in vivo at high doses while the low dose showed no effect. This might be explained by the breakdown of the drugs (metabolism). The authors need to show the plasma concentration of the drug during the treatment
· What was the effect of the treatment on other organs like the liver and kidney? Is there any side effect of consuming that high concentration of the drug?
· The author should try to test the effect of Anserine after PE-induced hypertrophy to show it is a treatment effect rather than a protective effect as he tested the effect before PE
Minor comments
1) Data should be represented as mean ± SD instead of SEM
2) The data should be represented as dots.
3) What concentration did he use in figure 3a,b
Author Response
Response to Reviewer 1,
Comment:
The authors discussed a hot research area which is the treatment of cardiac hypertrophy. they suggested that Anserine would be a potential treatment for pressure overload-induced cardiac hypertrophy.
The authors nicely presented their data and showed that Anserine prevented PE-induced hypertrophy in vitro and Marine active which contains Anserine as a main component reduced the effect of TAC-induced hypertrophy. but the results did not indicate whether is it the effect of Anserine alone or the combination of the 3 compounds. the authors tried to discuss this issue but here are some ideas that might help answer this question
Response:
Thank you very much for your helpful critique of our manuscript. In response to your comments, we have performed some experiments and provided the data in the revised manuscript.
Comment 1:
1) test the effect of different concentrations of carnosine, and histidine on PE-induced hypertrophy in vitro alone and a combination of the most effective doses.
Response 1:
Thank you for your valuable point. In this study, we investigated the inhibitory effects of anserine (10 mM) and L-histidine (10.6 mM) on cardiomyocyte hypertrophy. We found that anserine but not L-histidine inhibited PE-induced cardiomyocyte hypertrophy (Figure 4A, 4B). We did not determine anserine, carnosine, and L-histidine combination therapy on cardiomyocyte hypertrophy. We investigated the structure-activity relationship among the components of imidazole peptides. Although only 10 mM anserine prevented the reduction of PE-induced increase in cell surface area in cardiomyocytes, carnosine, β-alanine, and 1-methyl-histidine could not suppress this increase. These results implied that only anserine among the imidazole peptides had anti-hypertrophic effects in cardiomyocytes. Thus, the methyl group at position 1 of nitrogen and peptide bond with β-alanine in anserine were required for its anti-hypertrophic effects. In future research, we would like to clarify the structure-activity relationship study between the inhibition of p300-HAT activity and cardiomyocyte hypertrophy.
Comment 2:
2) Test the effect of Anserine alone in vivo instead of Marine.
Response 2:
Thank you for your valuable suggestion. In this study, we used Marine Active®, which contained a high amount of anserine (38%) for the in vivo experiment. As your suggestion, we understand that we should investigate the effect of anserine alone on the TAC-induced development of heart failure in vivo. Because purified anserine is very expensive compared to carnosine, we performed in vivo experiment using Marine Active® instead of anserine in this study. Our recent data found that anserine, but not carnosine, exhibited p300-HAT inhibitory activity in vitro and anti-hypertrophic effects in cardiomyocytes. Based on our findings, we will try to perform the comparison study of anserine alone and carnosine alone in vivo experiments in the future.
Comment 3:
The authors showed that Marine protected the heart against TAC-induced hypertrophy in vivo at high doses while the low dose showed no effect. This might be explained by the breakdown of the drugs (metabolism). The authors need to show the plasma concentration of the drug during the treatment
Response 3:
Thank you for the precious insight. It is not well understood how the metabolism of anserine is changed in pathological models, especially in cardiac diseases. We could not measure the concentration of serum anserine levels. However, to clarify whether oral administration of Marine Active containing anserine exhibits pharmacological activity in hearts, we measured the cardiac anserine levels in the heart tissue in each group according to the method of Shigemura, et al [1]. Chronic pressure overload reduced the anserine content in heart tissue compared to sham surgery. A high dose of Marine Active significantly increased cardiac anserine content compared to the TAC group with vehicle treatment. The amount of anserine in low doses of the Marine Active group was rarely increased. This finding supported that the high dose, but not low of the Marine Active group exerted anserine's therapeutic effect on the development of heart failure in vivo.
Interestingly, chronic pressure overload, such as TAC surgery, reduced cardiac anserine levels. Anserine is synthesized from carnosine by carnosine-N-methyltransferase enzyme (CARNMT1) and produced by transferring a methyl group onto carnosine [2]. Drozak et al. showed that mouse, human, and chicken carnosine synthase 1 (CARNS1) can synthesize anserine from β-alanine and 1-methyl-histidine [3]. Carnosine dipeptidase 1 and 2 (CNDP1 and CNDP2) can degrade carnosine and anserine [4]. Recently, anserine is present in the human cardiac muscle [5]. Anserine has been poorly studied compared to its analog, carnosine, and its biological relevance, especially in the heart, is largely unknown. Our recent study showed anserine's protective effect on cardiac hypertrophy and systolic dysfunction. The abnormality of anserine metabolism may be involved in the development of cardiac hypertrophy and heart failure.
Based on these findings, we added the data of cardiac anserine content in Figure 8 and mentioned them in the Results, Discussion, and Materials and Methods sections of the revised manuscript as follows:
Line 242-250,
“2.8 Oral administration of Marine Active® increased the content of anserine in heart tissue
To clarify whether the administration of Marine Active® accumulated anserine in the heart, we measured cardiac anserine content using the heart tissue of each group. Chronic pressure overload reduced the anserine content in heart tissue compared to sham surgery. A high dose of Marine Active® significantly increased cardiac anserine content compared to the TAC group with vehicle treatment. The amount of anserine in low doses of the Marine Active® group was rarely increased. This finding supported that the high dose, but not low, of the Marine Active group exerted the therapeutic effect of anserine on the development of heart failure in vivo.”
Line 308-344,
" Our collaborators showed that, when humans were orally administered with either anserine alone (2.0 g / 60 kg) or Marine Active® (19.4 g / 60 kg, providing anserine at 2.0 g / 60 kg) dissolved in water, there were no significant differences in the kinetic parameters AUC0-4, tmax, t1/2, or Cmax [32]. This indicates that the intestinal absorption and blood clearance of anserine are largely unaffected by other factors such as carnosine, other amino acids, proteins, and dextrin [32]. In this study, we detected the anserine content in heart tissue and found that a high dose of Marine Active® significantly increased the cardiac level of anserine (Figure 8C). Thus, as Marine Active® has the same effect as anserine alone, we speculate that cardiac accumulated anserine is the main factor in the inhibitory effect of Marine Active® on cardiac hypertrophy and cardiac dysfunction.
Interestingly, chronic pressure overload, such as TAC surgery, induced the reduction of cardiac anserine levels. Anserine is synthesized from carnosine by carnosine‐N‐methyltransferase enzyme (CARNMT1) and produced by transferring a methyl group onto carnosine [33]. Drozak, et al. showed that mouse, human, and chicken carnosine synthase 1 (CARNS1) can synthesize anserine from β-alanine and 1-methyl-histidine [34]. Carnosine dipeptidase 1 and 2 (CNDP1 and CNDP2) can degrade carnosine and anserine [35]. Recently, anserine is present in the human cardiac muscle [36]. Anserine has been poorly studied compared to its analog, carnosine, and its biological relevance, especially in the heart, is largely unknown. Our recent study showed anserine's protective effect on cardiac hypertrophy and systolic dysfunction. Since the abnormality of anserine metabolism may be involved in the development of cardiac hypertrophy and heart failure, it requires further analysis in the future.
In the present study, Marine Active® inhibited hypertrophy in primary cultured cardiomyocytes significantly more than anserine alone (Figure 3), suggesting that components other than anserine may affect anti-hypertrophic effects. Similarly, in the in vitro HAT assay, the IC50 of Marine Active® was 2.70 mM in terms of anserine, while that of an-serine alone was 1.87 mM. There was almost no difference in IC50 between anserine and Marine Active®. Marine Active® is made from skipjack and yellowfin tuna extracts and contains 38% anserine, 26.2% L-histidine, 3.4% carnosine, 7% other amino acids, and 18.4% protein. The equal amounts of L-histidine alone and carnosine alone did not inhibit the HAT activity of p300 in vitro. Some intercellular peptidases, such as CNDP2, hydrolyze anserine, carnosine, and dipeptides containing histidine [35]. It is possible that the carnosine and histidine in Marine Active® competitively affect CNDP2 activity and attenuate the degradation of anserine, resulting in the increased hypertrophic effect of Marine Active® over anserine. The structure-activity relationship analysis of anserine for HAT-inhibitory and anti-hypertrophic effects is necessary to understand the molecular mechanism of anserine in the future study. “
Line 354-362,
“6.1 Materials
Anserine (Figure 1), carnosine, L-histidine, 1-methyl-histidine, β-alanine, phenylephrine, acetonitrile, trifluoroacetic acid, trichloroacetic acid, aminoadipic acid, triethylamine, and phenyl isothiocyanate were purchased from Fujifilm Wako Pure Chemical (Osaka, Japan). Hydrophilic Nylon Syringe Filters (0.45 μm) was purchased from Hamach Scientific (Shaanxi, China). Anserine, carnosine, L-histidine, 1-methyl-histidine, and β-alanine were dissolved in distilled water and filtered before used. Marine Active® was provided by Yaizu Suisankagaku Industry (Shizuoka, Japan) and stored at 4°C. Marine Active® contains 38% anserine, 26.2% L-histidine, 3.4% carnosine, and 32.4% others.”
Line 455-470,
“6.12 Measurement of anserine content in heart tissue
The anserine content was measured in heart tissue according to the method of Shigemura, et al [45][1] with some modification. In brief, heart tissue (100 mg) was added in phosphate-buffered saline and homogenized using Micro SmashTM MS-100R at 5,000 rpm, 4oC, 80 sec. After centrifugation at 12,000 g, 4oC, 10 min, the supernatant was added trichloroacetic acid (5%), vortexed, centrifugated at 12,000 g, 4oC, 10 min, and filtrated through 0.45 μm hydrophilic nylon membrane. Aminoadipic acid was used as an internal control. Twenty μL of this solution was lyophilized. After phenylthiocarbamylation, the sample was dissolved in solvent A (0.01% trifluoroacetic acid pH 3.0) and solvent B (60% acetonitrile) (5:15, v/v), filtered, and analyzed using an HPLC system and Wakopak Wakosil-PTC (4.0 × 250 mm) column (Fujifilm Wako Pure Chemical, Osaka, Japan) at a flow rate of 0.5 mL/min. The column was equilibrated with 15% solvent B. The gradient profile was as follows: 0-30 min, 15-75% B; 30-35 min, 75-100 B; 35-40 min, 100% B; 40-50 min, 15% B. The column was maintained at 45oC. The absorbance at 254 nm was monitored. The anserine content was calculated using anserine standards, a calibration curve was constructed over a concentration range of 0.1 nM to 10 nM.”
Comment 4:
What was the effect of the treatment on other organs like the liver and kidney? Is there any side effect of consuming that high concentration of the drug?
Response 4:
Thank you for your comment. As you pointed out, it is crucial to determine whether Anserine has any side effects. To investigate the Morphological analysis of the liver and kidney with the treatment of Marine Active®, Formalin-fixed liver and kidney sections were stained with hematoxylin and eosin. We confirmed no effect on liver and kidney tissues with the treatment of a high dose of Marine Active®. The differences in tissue weights and morphology of liver and kidney in each group were not observed (Supplemental Figure and Table). Based on these results, it is likely that Anserine has no side effects.
We added a supplemental figure and table and mentioned it in the revised manuscript.
Line 168-185,
“2.4. Marine Active® suppressed transverse aortic constriction (TAC)-induced systolic dysfunction
The TAC mouse model was then used to investigate whether anserine suppresses the development of heart failure in vivo. One day after TAC surgery, the mice were allocated to one of three groups: Marine Active® low dose (anserine: 60 mg/kg/day), Marine Active® high dose (anserine: 200 mg/kg/day), or vehicle (0.5% CMC-Na in saline) as a control. Daily oral administration was continued for 8 weeks. Representative echocardiographic images are displayed in Figure 5A, and the echocardiographic and hemodynamic parameters are presented in Table 1. Eight weeks after treatment, the vehicle group demonstrated significantly decreased left ventricular fractional shortening (FS) and increased posterior wall theickness (PWd), and interventricular septum thickness in diastole (IVSD). These changes were suppressed by high dose of Marine Active® treatment. Pressure-overload induced cardiac hypertrophy (Figure 5B) and also displayed significant increases in heart weight-to-body weight (HW/BW, Figure 5C) and heart weight-to-tibia length (HW/TL, Figure 5D) ratios in the vehicle group. These increases were also attenuated by a high dose of Marine Active® treatment. There were no differences in liver and kidney weight-to-TL among each group (Supplemental Figure 1). These results indicate that high dose of Marine Active® significantly suppresses cardiac hypertrophy and prevents systolic dysfunction in the TAC mouse model. ”
Line 202-212
“2.5. Marine Active® suppressed TAC-induced cardiac hypertrophy
To investigate whether Marine Active® suppresses TAC-induced cardiac hypertrophy in mice, hematoxylin and eosin (HE) staining was performed (Figure 6A). High dose of Marine Active® treatment significantly, but not completely, suppressed TAC-induced cardiomyocyte hypertrophy (Figure 6B). In addition, the liver and kidney tissues were stained with HE staining. The abnormalities of these tissues were not observed (Supplemental Figure 2). Next, to investigate whether Marine Active® suppresses TAC-induced hypertrophy-related gene transcription, quantitative RT-PCR was performed. High dose of Marine Active® treatment significantly suppressed the transcription of ANF and BNP (Figure 6C, D). These findings suggest that high dose of Marine Active® inhibits the TAC-induced cardiac hypertrophic response in mice. ”
Comment 5:
The author should try to test the effect of Anserine after PE-induced hypertrophy to show it is a treatment effect rather than a protective effect as he tested the effect before PE
Response 5:
Thank you for your suggestion. There is a standard protocol for the treatment method with a compound before PE stimulation [6–11]. Indeed, it would be very interesting to determine whether the compound exerts a therapeutic effect, and we would like to investigate this in the future.
Comment 6:
Minor comments
1) Data should be represented as mean ± SD instead of SEM
2) The data should be represented as dots.
3) What concentration did he use in figure 3a,b
Response 6:
Thank you for your valuable comments. We represented all data as the mean ± SD. We represented the data as dots in animal experiments (Figure 5-8). The concentration of anserine was described in Figures 3A and 3B.
Reference
- Shigemura, Y.; Iwasaki, Y.; Sato, Y.; Kato, T.; Seko, T.; Ishihara, K. Detection of Balenine in Mouse Plasma after Administration of Opah-Derived Balenine by HPLC with PITC Pre-Column Derivatization. Foods 2022, 11, 590, doi:10.3390/foods11040590.
- Kwiatkowski, S.; Kiersztan, A.; Drozak, J. Biosynthesis of Carnosine and Related Dipeptides in Vertebrates. Curr Protein Pept Sci 2018, 19, 771–789, doi:10.2174/1389203719666180226155657.
- Drozak, J.; Veiga-da-Cunha, M.; Vertommen, D.; Stroobant, V.; Van Schaftingen, E. Molecular Identification of Carnosine Synthase as ATP-Grasp Domain-Containing Protein 1 (ATPGD1). Journal of Biological Chemistry 2010, 285, 9346–9356, doi:10.1074/jbc.M109.095505.
- Bellia, F.; Vecchio, G.; Rizzarelli, E. Carnosinases, Their Substrates and Diseases. Molecules 2014, 19, 2299–2329, doi:10.3390/molecules19022299.
- de Souza Gonçalves, L.; Pereira, W.R.; da Silva, R.P.; Yamaguchi, G.C.; Carvalho, V.H.; Vargas, B.S.; Jensen, L.; de Medeiros, M.H.G.; Roschel, H.; Artioli, G.G. Anserine Is Expressed in Human Cardiac and Skeletal Muscles. Physiol Rep 2023, 11, e15833, doi:10.14814/phy2.15833.
- Ritterhoff, J.; Young, S.; Villet, O.; Shao, D.; Neto, F.C.; Bettcher, L.F.; Hsu, Y.-W.A.; Kolwicz, S.C.; Raftery, D.; Tian, R. Metabolic Remodeling Promotes Cardiac Hypertrophy by Directing Glucose to Aspartate Biosynthesis. Circ Res 2020, 126, 182–196, doi:10.1161/CIRCRESAHA.119.315483.
- Boluyt, M.O.; Zheng, J.S.; Younes, A.; Long, X.; O’Neill, L.; Silverman, H.; Lakatta, E.G.; Crow, M.T. Rapamycin Inhibits Alpha 1-Adrenergic Receptor-Stimulated Cardiac Myocyte Hypertrophy but Not Activation of Hypertrophy-Associated Genes. Evidence for Involvement of P70 S6 Kinase. Circ Res 1997, 81, 176–186, doi:10.1161/01.res.81.2.176.
- Tokudome, T.; Horio, T.; Kishimoto, I.; Soeki, T.; Mori, K.; Kawano, Y.; Kohno, M.; Garbers, D.L.; Nakao, K.; Kangawa, K. Calcineurin-Nuclear Factor of Activated T Cells Pathway-Dependent Cardiac Remodeling in Mice Deficient in Guanylyl Cyclase A, a Receptor for Atrial and Brain Natriuretic Peptides. Circulation 2005, 111, 3095–3104, doi:10.1161/CIRCULATIONAHA.104.510594.
- Planavila, A.; Rodríguez-Calvo, R.; Jové, M.; Michalik, L.; Wahli, W.; Laguna, J.C.; Vázquez-Carrera, M. Peroxisome Proliferator-Activated Receptor Beta/Delta Activation Inhibits Hypertrophy in Neonatal Rat Cardiomyocytes. Cardiovasc Res 2005, 65, 832–841, doi:10.1016/j.cardiores.2004.11.011.
- Markou, T.; Hadzopoulou-Cladaras, M.; Lazou, A. Phenylephrine Induces Activation of CREB in Adult Rat Cardiac Myocytes through MSK1 and PKA Signaling Pathways. J Mol Cell Cardiol 2004, 37, 1001–1011, doi:10.1016/j.yjmcc.2004.08.002.
- Pu, W.T.; Ma, Q.; Izumo, S. NFAT Transcription Factors Are Critical Survival Factors That Inhibit Cardiomyocyte Apoptosis during Phenylephrine Stimulation in Vitro. Circ Res 2003, 92, 725–731, doi:10.1161/01.RES.0000069211.82346.46.

Reviewer 2 Report
Comments and Suggestions for Authors
Dear Authors,
Your original paper, which demonstrates anserine's potential in mitigating cardiac hypertrophy and systolic dysfunction induced by pressure overload, suggests its viability as a pharmacological agent for heart failure therapy. It is interesting and of significant relevance to cardiovascular research. Although the manuscript is well-written, there are a few issues that must be addressed.
- Table 1:
- First line, second column: Consider changing "forword" to "forward."
- Lines 239-240:
- Revise the statement as its present form is confounding. Additionally, there is one end bracket without a corresponding start bracket.
- Figure 4:
- There is no correlation between images and explanations (panels C and D). Please proceed with adequate changes and refer to the text as well.
- Figure 5:
- There is no correlation between images and explanations (panels B and D). Please proceed with adequate changes and refer to the text as well.
- Table 2:
- The title should be above the table, not beneath it (similar to Table 1).
- Reconsider values in the headings of columns 4 and 5, respectively, as both state "high dose Marine Active" (which is not true according to the study method).
Thank you for your attention to these matters. Your cooperation in addressing these issues will enhance the overall quality of your manuscript.
Best regards
Author Response
Response to Review 2,
Comment:
Your original paper, which demonstrates anserine's potential in mitigating cardiac hypertrophy and systolic dysfunction induced by pressure overload, suggests its viability as a pharmacological agent for heart failure therapy. It is interesting and of significant relevance to cardiovascular research. Although the manuscript is well-written, there are a few issues that must be addressed.
Response:
Thank you very much for your valuable comments on our manuscript. We have revised our manuscript according to your suggestion:
Comment 1:
Table 1:
First line, second column: Consider changing "forword" to "forward."
Response 1:
Thank you for your point. We corrected it in the revised manuscript.
Lines 239-240:
Revise the statement as its present form is confounding. Additionally, there is one end bracket without a corresponding start bracket.
There is no correlation between images and explanations (panels C and D). Please proceed with adequate changes and refer to the text as well.
Response 2-3:
Thank you for your points. We deleted the one-end bracket and corrected the explanations in the revised manuscript.
Line 146-153,
“ We next performed an in vitro p300-HAT assay using 6.3 mg/mL Marine Active®, 10.6 mM L-histidine, and 1.0 mM carnosine. Acetylation of histone H3K9 was significantly inhibited by 6.3 mg/mL Marine Active® but not by 10.6 mM L-histidine, and 0.9 mM carnosine (Figure 4C). The IC50 value of the p300-HAT inhibitory effect of Marine Active® was calculated using the sigmoid dose–response curves with variable slopes, resulting in an estimated value of 4.64 mg/mL (2.70 mM anserine) (Figure 4D). These results suggest that Marine Active® inhibits p300-HAT activity and cardiomyocyte hypertrophy to the same extent as anserine. ”
Comment 4:
Figure 5:
There is no correlation between images and explanations (panels B and D). Please proceed with adequate changes and refer to the text as well.
Response 4:
Thank you for your comment. We corrected the figure orders and explanations in the revised manuscript.
Line 168-185,
“2.4. Marine Active® suppressed transverse aortic constriction (TAC)-induced systolic dysfunction
The TAC mouse model was then used to investigate whether anserine suppresses the development of heart failure in vivo. One day after TAC surgery, the mice were allocated to one of three groups: Marine Active® low dose (anserine: 60 mg/kg/day), Marine Active® high dose (anserine: 200 mg/kg/day), or vehicle (0.5% CMC-Na in saline) as a control. Daily oral administration was continued for 8 weeks. Representative echocardiographic images are displayed in Figure 5A, and the echocardiographic and hemodynamic parameters are presented in Table 1. Eight weeks after treatment, the vehicle group demonstrated significantly decreased left ventricular fractional shortening (FS) and increased posterior wall thickness (PWd) and interventricular septum thickness in diastole (IVSD). These changes were suppressed by high dose of Marine Active® treatment. Pressure-overload induced cardiac hypertrophy (Figure 5B) and also displayed significant increases in heart weight-to-body weight (HW/BW, Figure 5C) and heart weight-to-tibia length (HW/TL, Figure 5D) ratios in the vehicle group. These increases were also attenuated by a high dose of Marine Active® treatment. There were no differences in liver and kidney weight-to-TL among each group (Supplemental Figure 1). These results indicate that high dose of Marine Active® significantly suppresses cardiac hypertrophy and prevents systolic dysfunction in the TAC mouse model. ”
Line 197-201,
“Figure 5. TAC-induced systolic dysfunction was suppressed by Marine Active®
The results of echocardiography 8 weeks after treatment. (A) Representative M-mode echocardiograms of mouse heart. (B) Representative images of whole hearts from the mice. Scale bar: 5 mm. (C) The results of HW/BW. Data are presented as the mean ± SD (n = 6). (D) The results of HW/TL. Data are presented as the mean ± SD (n = 6).”
Comment 5:
Table 2:
The title should be above the table, not beneath it (similar to Table 1).
Reconsider values in the headings of columns 4 and 5, respectively, as both state "high dose Marine Active" (which is not true according to the study method).
Response 5:
Thank you for your point. We corrected it in the revised manuscript.

Round 2
Reviewer 1 Report
Comments and Suggestions for Authors
accept.
the authors addressed all the comments